# A Novel Ultra-Low Power Consumption Electromagnetic Actuator Based on Potential Magnetic Energy: Theoretical and Finite Element Analysis

**M. Albertos-Cabanas, D. Lopez-Pascual** **, I. Valiente-Blanco \*, G. Villalba-Alumbreros and M. Fernandez-Munoz**

Mechanical Engineering Area, Universidad de Alcalá, Ctra. Madrid-Barcelona, km 33.600, 28805 Alcalá de Henares, Spain
\* Correspondence: i.valiente@uah.es

**Abstract:** A novel concept of a rotary electromagnetic actuator for positioning with ultra-low power consumption is presented. The device is based on harnessing potential magnetic energy stored between permanent magnets facing each other with opposing magnetization polarities. When combined with an active electromagnetic control and passive stabilization system, the rotor of the device can switch between stable equilibrium positions in a fast way with a minimal fraction of the power and energy consumption of a traditional electromagnetic actuator. In this paper, a theoretical model, supported by finite element analysis results, is presented. The actuator has been designed in detail to operate as an optical filter wheel actuator. Calculations demonstrate that the device has the potential to provide a power-consumption saving of up to 86.6% and an energy consumption reduction of up to 58.6% with respect to a traditional filter wheel actuator.

**Keywords:** electromagnetic actuator; efficiency; energy consumption; potential energy

## 1. Introduction

Energy consumption of actuation systems is an engineering problem of particular relevance due to the constant need to increase the energy efficiency and sustainability of production processes and final user applications. Electromagnetic actuators are used in various applications where fast and accurate positioning is required. Examples of such applications are process feeding, serial manufacturing systems, and tool change systems for machining centers [1], robotic manipulators [2], and actuators for biomedical applications [3,4]. Other significant applications require actuators with high dynamic performance, capable of providing high accelerations and powerful braking or damping systems [5–7], which frequently have high associated costs and energy and power requirements.

An application of particular interest for the present development is filter wheel mechanisms found in numerous optical applications. A filter wheel is a rotating device that carries a series of optical filters for different wavelengths. Optical filters can be designed to transmit, block, or reflect light over any wavelength range from UV to IR [8]. Filter wheels are a very common piece of equipment in optical laboratories and are also frequently used in ground and space telescopes. In the case of flight equipment actuation with low power consumption, these are of particular relevance due to the reduced power budgets on board satellites [9,10].

In particular, the low power consumption requirement is critical in missions that operate in cryogenic environments at very low temperatures [11–13]. A good example of such missions is the James Webb Space Telescope [14]. On this telescope, there are various filter wheels with different functionalities. The NIRCam instrument has a double-wheel assembly with 12 filters each, consisting of a pupil wheel and a filter wheel placed in parallel. The collimator light passes first through the active element of the pupil wheel and then



through the active element of the filter wheel. The two wheels can rotate independently to select specific combinations of the optical elements that define the observation modes of the NIRISS [15]. On the other hand, the MIRI instrument contains a filter wheel that carries 18 selectable optical elements: narrowband and broadband filters, a prism, and four chronographic masks. The two nearly identical grid/dichroic wheels carry combinations of grids and dichroics with only three positions for each wheel [16]. Table 1 summarizes the main performance of the DC electromagnetic actuators incorporated into the mentioned filter wheels of the James Webb Space Telescope.

**Table 1.** Performance of the engines of the MIRI and NIRCam instruments of the James Webb telescope [15–17].

| Instrument | FWA DM Cryo 84 | NIRCam Actuator |
|---|---|---|
| **Parameter** | **Value** | **Value** |
| Actuator diameter | 96 mm | 91.4 |
| Actuator length | 22 mm | n/a |
| Rotor inertia | 0.25 gm$^2$ | n/a |
| Mass | 0.53 kg | n/a |
| Number of magnets | 24 | n/a |
| Number of coils | 12 | n/a |
| Resistance winding (293 K) | 370 ohms | 82 ohms |
| Winding inductance | 19 mH | n/a |
| Maximum current | 250 mA | 130 mA |
| Maximum voltage | 40 V | n/a |
| Consumption max. (293 K) | 23 watts | 3.3 watts |
| Temperature range | 4.2–300 K | 6–300 K |
| Acting torque | 200 mNm | 149.5 mNm |
| Change time of filter | 500 ms | n/a |
| Energy consumed for position change (estimated) | 11.5 J | n/a |

Power consumption can be significantly reduced in both cases when instruments operate under cryogenic conditions due to reduced winding ohmic resistance [18]. However, despite the optimization achieved by some of the aforementioned developments, their operational concept prevents the optimization of the electrical consumption of these devices since a large amount of energy is needed to accelerate and brake the payload to meet the strict requirements of filter change.

This research paper presents a novel rotary electromagnetic actuator based on the use of potential magnetic energy to minimize the power and electrical consumption of the device. Potential magnetic energy can be stored by facing permanent magnets with opposite polarities [19]. This approach has been previously explored with good results for the optimization of energy harvesting devices [20,21]. The potential energy is released in a controlled manner by an active actuation system, minimizing the time of position change and minimizing the energy consumed by the system. The power requirements of the actuator are also reduced, which may also reduce the size, cost, and weight of the associated power electronics.

In this paper, the theoretical framework for the operation of such a device and its detailed design as an actuator for a filter wheel application is presented. The estimated performance of the device has been calculated using a theoretical model supported by finite element model (FEM) simulations.

## 2. Principle of Operation

The proposed device is composed of three main elements:

1. A rotor, mainly composed of permanent magnets.
2. A stator, mainly composed of a second set of permanent magnets. Their magnetization directions are coincident with those of the stator and axis of rotation of the device. However, the sense of magnetization of these permanent magnets is opposite to that of rotor magnets.
3. A stabilization and actuation system, placed in the stator, mainly composed of a series of ferromagnetic elements that stabilize the equilibrium positions of the actuator and a set of coils or electromagnets that allow the actuator to rotate in both directions.

Figure 1 schematizes these elements in a section view of a radial actuator with four equilibrium positions based on this novel concept. Because the permanent magnets of the rotor and the stator present opposing polarities, a large amount of potential magnetic energy is stored in the device when the rotor and stator magnets are close to each other. However, without the aid of soft magnetic materials for stabilization in the stator (stabilizer), the equilibrium positions $n$ = 1, 2, 3, and 4 shown in Figure 1 would be positions of unstable equilibrium. Consequently, under the presence of any external disturbance, the rotor would uncontrollably abandon these equilibrium positions. The problem is solved by incorporating a set of parts made of soft magnetic materials in the surroundings of the equilibrium positions. When properly dimensioned, these parts generate a local magnetic potential that stabilizes the rotor in positions 1 to 4. Then, the rotor will remain in these positions, even if there are small external disturbances present.

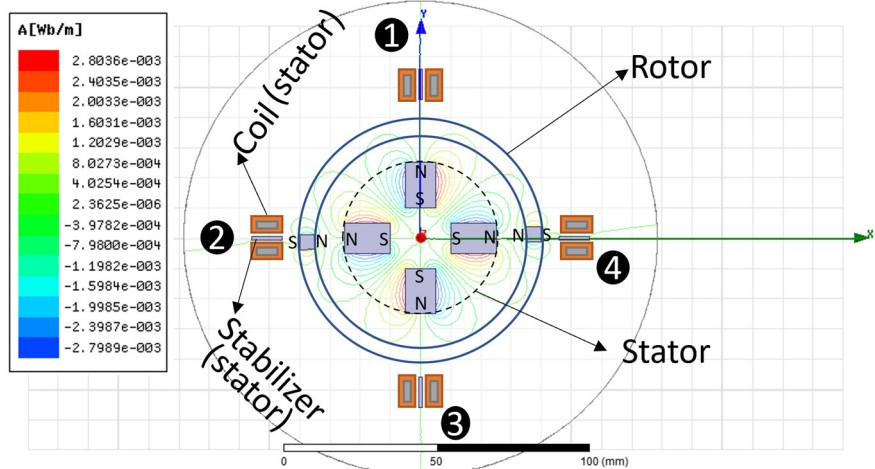

**Figure 1.** Main elements and stable equilibrium positions of the radial actuator.

Finally, the stored potential magnetic energy can be released and transformed into kinetic energy by a set of actuation coils in the stator of the actuator. By controlling the current level and current flow direction, the rotor is allowed to move in both directions and change from one equilibrium position to another. To do so, only a minimal fraction of energy is required to overcome the potential well, saving a significant amount of energy in each position change. If any other energy dissipation phenomenon is present in the system, for example, eddy currents dissipation or friction in the bearing, extra energy shall be provided by the actuation system to be able to change from one equilibrium position to another.

Figure 2 illustrates the rotor potential energy vs. the angular position of the actuator shown in Figure 1.

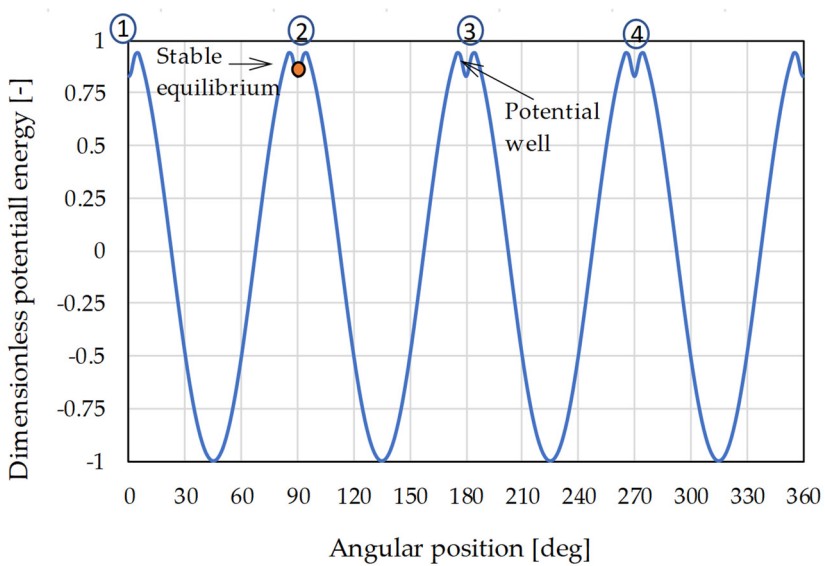

**Figure 2.** View in plan and principle of operation.

The principle of harnessing the potential energy stored between magnets in repulsion is what allows for a significant reduction in the requirements of power and energy of actuation. It is no longer necessary to supply the energy to accelerate and brake the rotor, but simply the energy needed to get the rotor out of its stable equilibrium position, which as will be demonstrated, is much lower than the previous one.

### 3. Theoretical Model

From a theoretical point of view, it can be established, based on the principle of conservation of the angular momentum (*L*) of the rotor, that:

$$\frac{\partial \vec{L}}{dt} = [I]\cdot\vec{\alpha} + \vec{\omega} \times \left([I]\cdot\vec{\omega}\right) = \sum \vec{M} \tag{1}$$

where $[I]$ is the inertia matrix of the rotor, $\vec{\alpha}$ its angular acceleration, $\vec{\omega}$ its angular velocity, $\vec{M}$ any external moment applied, and *t* is time.

As long as rotation takes place around the main axis of inertia of the rotor (such as the *z*-axis), the above equation can be reduced to:

$$\sum M_z = I_z\cdot\alpha_z \tag{2}$$

Equation (2) can be modified using a variable change, such as:

$$\sum M_z = I_z\cdot\frac{d\omega_z}{\partial\theta}\cdot\frac{\partial\theta}{\partial t} \tag{3}$$

Or:

$$\sum M_z = I_z\cdot\frac{\partial\omega_z}{\partial\theta}\cdot\omega_z \tag{4}$$

where $\omega_z$ is the angular velocity of the rotor, and $\theta$ its angular position.

By operating and integrating the previous equation for a given displacement range of the rotor:

$$\int M_{z\_total}\cdot\partial\theta = I_z \int \omega_z\cdot\partial\omega_z \tag{5}$$

And finally, for the given angular displacement $\theta_f - \theta_i$, it can be said that:

$$\omega_{z\_f} = \sqrt{2 \cdot \frac{\int_{\theta_i}^{\theta_f} M_{z_{total}} \cdot \partial\theta}{I_z} + (\omega_{z\_i})^2} \tag{6}$$

where $\omega_{z\_f}$ is the final angular velocity of the device at the angular position $\theta_f$, $\omega_{z\_i}$ is the initial angular velocity for the starting position $\theta_i$, and $M_{z_{total}}$ is the summation of all external momentum applied to the rotor.

The external momentums applied to the rotor include the momentum exerted by the magnets arranged in the stator of the device, which mainly depends on the angular position of the rotor. In addition, other electromagnetic interactions (such as those induced by the ferromagnetic cores of the coils) exert an action on the rotor too. These interactions are extremely complex to calculate analytically; therefore, finite element software has been used to determine the dependence of the moments exerted on the rotor based on its angular position. Finally, any other external momentum, such as the bearings' friction torque, shall be included in the calculations.

From the angular velocity and torque profile of the rotor, a set of useful variables can be calculated. Indeed, the elapsed time can be approximated ($\Delta t$) between two angular positions very close to each other by applying finite differences, so that:

$$\Delta t = \frac{\Delta\theta}{\overline{\omega_z}} \tag{7}$$

where $\overline{\omega_z}$ is the average angular velocity between the two angular positions:

$$\overline{\omega_z} = \frac{\omega_{z\_f} + \omega_{z\_i}}{2} \tag{8}$$

Once the performance of the device in terms of torque, angular velocity, and position change time has been calculated, it is necessary to estimate the electrical power and its energy consumption. The instantaneous electrical power consumption of the actuation system can be calculated as:

$$W_{act} = I^2 \cdot R \tag{9}$$

where $I$ is the current through the winding and $R$ is the total resistance of the winding. As it will be mentioned later, only one actuation phase is required in this device; therefore, the total power consumption is the power consumption of the single-phase winding.

The resistance of the winding is determined by the section of the magnetic wire and its length. This resistance is dependent on temperature; therefore:

$$R = \rho \cdot (1 + \alpha(T - 20)) \cdot \frac{l}{A} \tag{10}$$

where $\rho$ is the resistivity of copper, $\rho = 1.7 \cdot 10-8$ *Ohm·m*, $\alpha$ is the temperature coefficient of copper resistivity $\alpha = 3.9 \cdot 10-3\,°C^{-1}$, $T$ is the temperature of the wire in $°C$, $A$ the wire section, and $l$ its length.

To obtain the potential power saving of the actuator, the previously calculated power consumption ($W_{act}$) is compared to what is called the apparent power consumption ($W_{ap}$) of the device. The latter represents the power consumption that a classical electromagnetic actuator should provide to reproduce the same torque vs. angular position characteristic curve, regardless of the electric motor configuration. The instantaneous apparent power consumption is calculated as:

$$W_{ap} = M_{z\_total} \cdot \omega_z \tag{11}$$

Then, the ideal ratio of power saving is defined considering the peak power apparent power consumption and the peak actuation power consumption, which do not consider any energy dissipation phenomena:

$$\eta_{W\_ideal}[\%] = \left[1 - \frac{W_{act\_max}}{W_{ap\_max}}\right] \cdot 100 \tag{12}$$

Note that the actuation system should be operative for only a short fraction of the time required for the rotor to leave the potential well. Therefore, the energy consumed for a change of position can be estimated via integration of the torque vs. angular position curve in the displacement range where the stabilization occurs.

$$E_{act} = \int_{\theta_{i\_stab}}^{\theta_{f\_stab}} |M_{z_{total}}| \cdot \partial\theta \tag{13}$$

where $\theta_{f\_stab}$ is the end of the stabilization zone and $\theta_{i}\_stab$ is the initial position of equilibrium.

On the other hand, the apparent energy consumed by a classical actuator could be obtained as:

$$E_{ap} = \int_{\theta_i}^{\theta_f} |M_{z_{total}}| \cdot \partial\theta \tag{14}$$

where, for a contiguous position change, $\theta_f$ is the final equilibrium position angular position and $\theta_i$ is the initial angular position.

Finally, the ideal energy-saving ratio is defined as:

$$\eta_{E\_ideal}[\%] = \left[1 - \frac{E_{act}}{E_{ap}}\right] \cdot 100 \tag{15}$$

However, in a realistic device, there will be other sources of energy dissipation such as magnetic losses or friction in the bearings that shall be considered. Therefore, Equations (12) and (15) can be easily modified to accommodate any additional energy dissipation phenomena as:

$$\eta_W[\%] = \left[1 - \frac{W_{act} + \overline{W}_{loss\_act}}{W_{ap} + \overline{W}_{loss\_ap}}\right] \cdot 100 \tag{16}$$

where $\eta_W$ is the power-saving ratio, $\overline{W}_{loss\_act}$ represents the average power loss induced by any physical phenomena in the actuator (for example, friction in the bearings or eddy currents dissipation), and $\overline{W}_{loss\_ap}$ is the average power loss induced in the classical electromagnetic actuator to be compared. For the sake of simplicity, we will consider the additional power losses of the novel and the classical actuator approximately equivalent; therefore, $\overline{W}_{loss\_act} \approx \overline{W}_{loss\_ap}$.

Finally:

$$\eta_E[\%] = \left[1 - \frac{E_{act} + \overline{E}_{loss\_act}}{E_{ap} + \overline{E}_{loss\_ap}}\right] \cdot 100 \tag{17}$$

where $\eta_E$ is the energy-saving ratio, $\overline{E}_{loss\_act}$ represents the energy loss induced by any physical phenomena in the actuator and $\overline{E}_{loss\_ap}$ is the energy loss induced in the classical electromagnetic actuator to be compared. For the sake of simplicity, we have considered the additional energy losses of the novel and the classical actuator approximately equivalent; therefore, $\overline{E}_{loss\_act} \approx E_{loss\_ap}$.

Furthermore, the presence of any energy dissipation phenomena would inexorably reduce the kinetic energy of the rotor, ultimately preventing it from achieving the next stable position. In order to overcome this limitation, as discussed in Section 2, the actuation system provides the extra energy required to compensate for those losses. By measuring the position and velocity of the rotor, a feedback control strategy could be potentially

defined in order to provide an accurate control of the positioning of the actuator. Although the definition of this control strategy is out of the scope of this paper, a set of Hall effect sensors, to be used with control purposes, have been foreseen in the design of the actuator in Section 4.

## 4. Mechanical and Electrical Design

A rotary actuator with eight equilibrium positions has been designed. The electromagnetic actuator was designed in an axial configuration. It is known, from classical electromagnetic and magneto-mechanical devices' design, that axial configurations typically provide superior torque density and specific actuation torque, therefore minimizing the device weight and volume [22]. Figure 3 shows the detailed design of the actuator and its main elements.

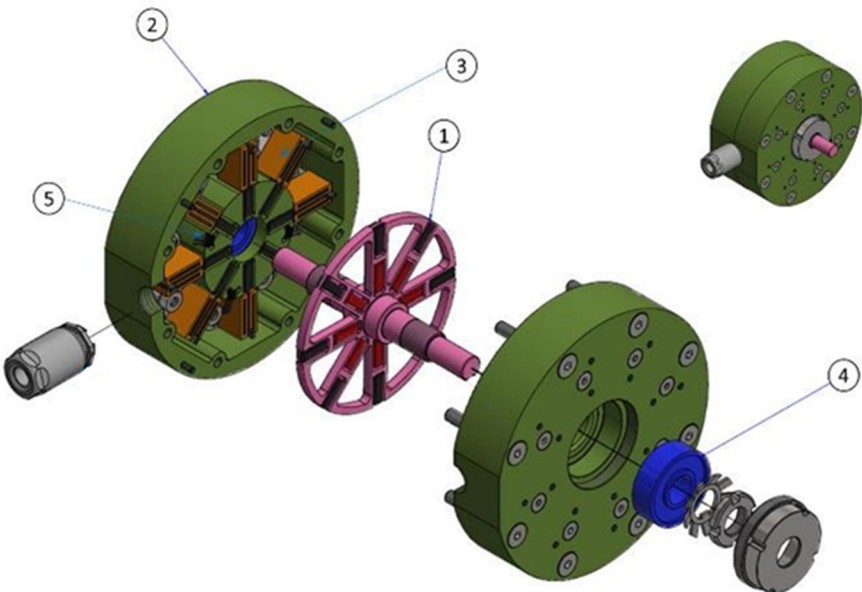

**Figure 3.** Detailed design of the VALMAC actuator. 1. Rotor, 2. Stator, 3. Coils, 4. Ball bearings, 5. Hall effect sensor.

The actuator measures 92 mm in diameter and has an estimated weight of 1.2 kg. The eight equilibrium positions are equidistant from each other by 45 degrees. The actuator is mainly composed of a rotor (1) and a stator (2). The rotor consists of an AISI 304 stainless steel wheel of 67 mm of outer diameter on which 16 permanent magnets of SmCo 30 are assembled, with a remanence value Br = 1.12 T and a coercivity Hc = 835 kA/m. The magnets' dimensions are $10 \times 10 \times 3$ mm and they are magnetized in the direction of the actuator shaft (z-axis) parallel to the magnet's thickness of 3 mm. A solid AISI 304 shaft measuring 8 mm in diameter and 82 mm in length mainly completes the rotor design. The moment of inertia of the rotor is calculated in $2.15 \cdot 10^{-5}$ kgm$^2$.

The stator (2) of the device is split into two cylindrical parts as shown in Figure 3. The airgap between the rotor and the stator is 0.5 mm. The actuator stator is mainly composed of a set of 16 permanent magnets of the same characteristics as the rotor magnets, assembled in two structural pieces of AISI 420 ferromagnetic stainless steel. The magnets of the stator, however, present an opposite magnetization direction with respect to the magnets of the rotor, generating the potential energy necessary for the actuation. In each equilibrium position, a stabilization trap made of AISI 1010 ferromagnetic steel is provided. This trap mainly generates the potential well required for the stabilization of the equilibrium positions and each of them measures $13 \times 20 \times 1$ mm. A total of 32 small ferromagnetic core coils (3) are placed in the stator. Each coil core is $13 \times 7.5 \times 1$ mm, and they are placed adjacent to both sides of each stabilization trap. Each coil is designed with 75 turns of magnetic wire AWG 28 with an estimated fill factor of 85%. The resistance of each coil

is estimated at 0.34 Ohm (at 70 °C) and its inductance at 0.45 μH. The actuation system only requires one phase of actuation. The magnetic field generated by coils is parallel to the magnetization direction of the permanent magnets (z-axis). The direction of motion is established by the current flow direction in the coils. Finally, the actuation winding is split into a nominal winding composed of 16 coils assembled in a series, and a secondary redundant winding composed of the remaining 16 coils. Redundant windings are required in space applications to improve the reliability of the system and to provide a solution in case of critical failure of one of the windings. The total resistance when both windings are operating at the same time is calculated at 11 Ohm (at 70 °C) and the total inductance at 14.5 μH. When the windings are operating in redundant mode (only one active), the total resistance is about 5.5 Ohm and the inductance 7.3 μH. Figure 4 shows a detailed view of the magnetic components in the device and their position.

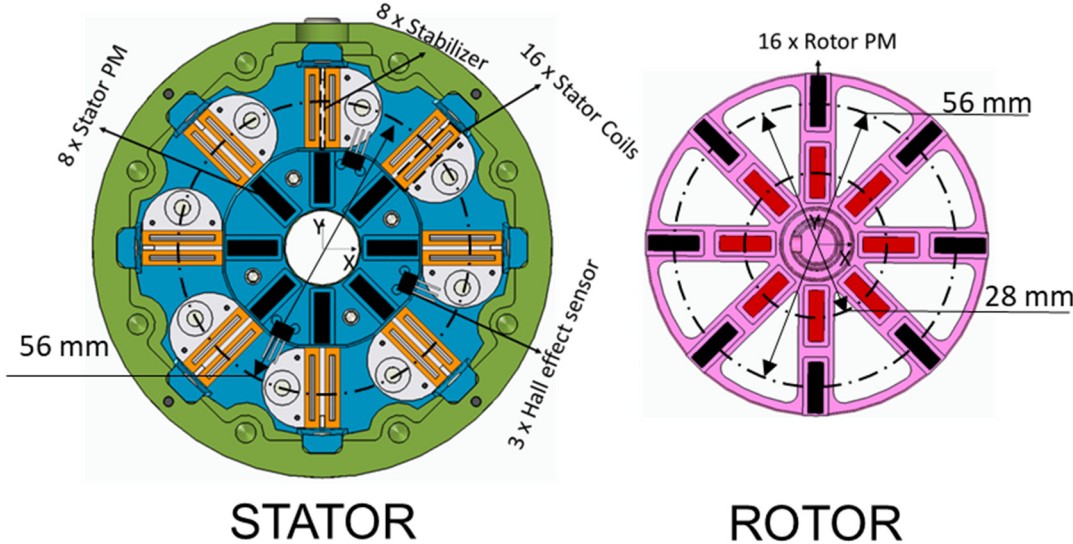

**Figure 4.** Half of the stator of the device (**left**) and rotor (**right**).

To allow rotation between the actuator and the rotor, two angular contact ball bearings (4) of 28 mm outer diameter and 8 mm thickness have been used. The installation of a series of Hall effect magnetic sensors (5) is also foreseen for position characterization of the rotor. An operational temperature range has been specified for the device between −0 °C and 70 °C. Table 2 summarizes the main constructive characteristics of the device.

**Table 2.** Device main characteristics.

| Parameter | Value |
|---|---|
| Actuator diameter | 92 mm |
| Actuator length | 40 mm |
| Rotor moment of inertia | $2.15 \cdot 10^{-5}$ kgm$^2$ |
| Actuator weight | 1.2 kg |
| Rotor: Number of magnets | 16 |
| Stator: Number of magnets | 16 |
| Permanent magnets material | SmCo 32 |
| Number of coils (single phase) | 32 (simultaneous), 16 (redundant) |
| Wire section | AWG 28 |
| Winding resistance at 70 °C (both windings) | 11 Ohms |
| Coil inductance (both windings) | 14.5 μH |

## 5. Finite Element Model (FEM)

The performance of the prototype is evaluated using two different FEMs that have been set up and solved using Ansys Electronics 2021 R1 software. In the first place, a magnetostatic simulation model has been set up. This FEM was used for the calculation of the torque vs. angular displacement function of the prototype and consists of a reduced system with the main magnetic elements of the actuator: coils, permanent magnets, and ferromagnetic yokes, but without bearings and other accessories. Figure 5 shows an image of the model.

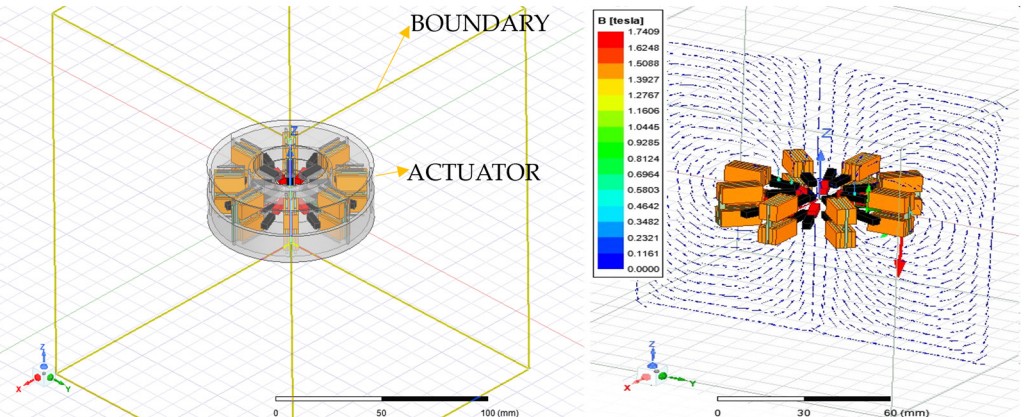

**Figure 5.** Magnetostatic FEM.

The model has been tuned with a maximum convergence error of 1%, with a typical number of approximately 300,000 elements. Solving the model provides the electromagnetic fields and, through post-processing techniques, the forces, and moments in the different components of the actuator. Figure 6 shows the airgap magnetic flux density at two rotor angular positions (angular position = 0 degrees and angular position = 22.5 degrees). Magnetic flux density is calculated in a circumference of radius (R) equivalent to the distance between the center of the device and the center of mass of the external PMs of the rotor (R = 28 mm) and located at half the distance between that PM surface and the stabilizer. Such a circumference is drawn in Figure 4 (right).

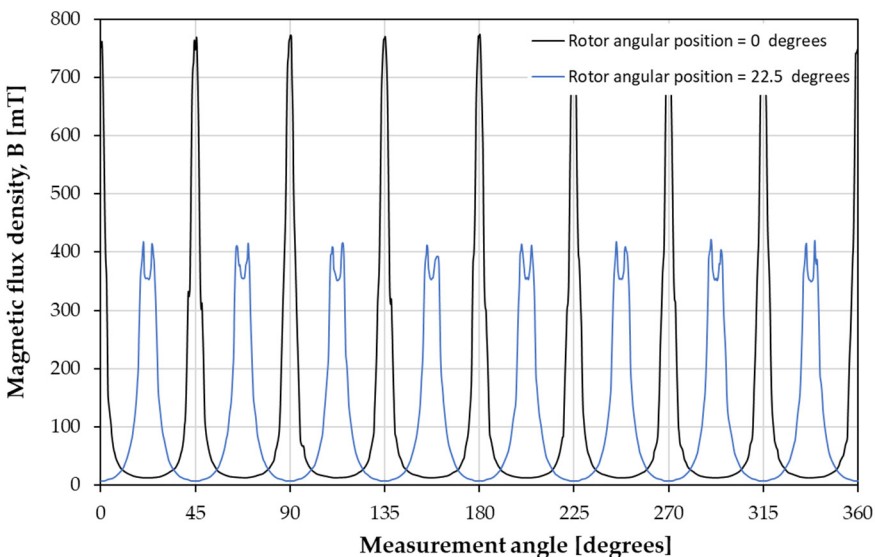

**Figure 6.** Magnetic flux density in the airgap for various positions of the rotor and position of measurement.

The maximum magnetic flux density is calculated in 0.78 T when the rotor is located in any of the generated equilibrium positions (angular position = 0+ $n \cdot 45$ degrees, being $0 \leq n \leq 7$).

In addition, the flux leakage is evaluated in a 2D cross section of the device in the XZ plane in Figure 4. Figure 7 shows the flux lines' distribution in two alternative positions of the rotor. In one of these positions, the rotor is located in a stable position (e.g., angular position = 0 degrees). In the other one, the rotor is located at half the angular distance between two adjacent stable positions (e.g., angular position = 22.5 degrees). Due to the symmetry in the device, only half of the cross section is represented.

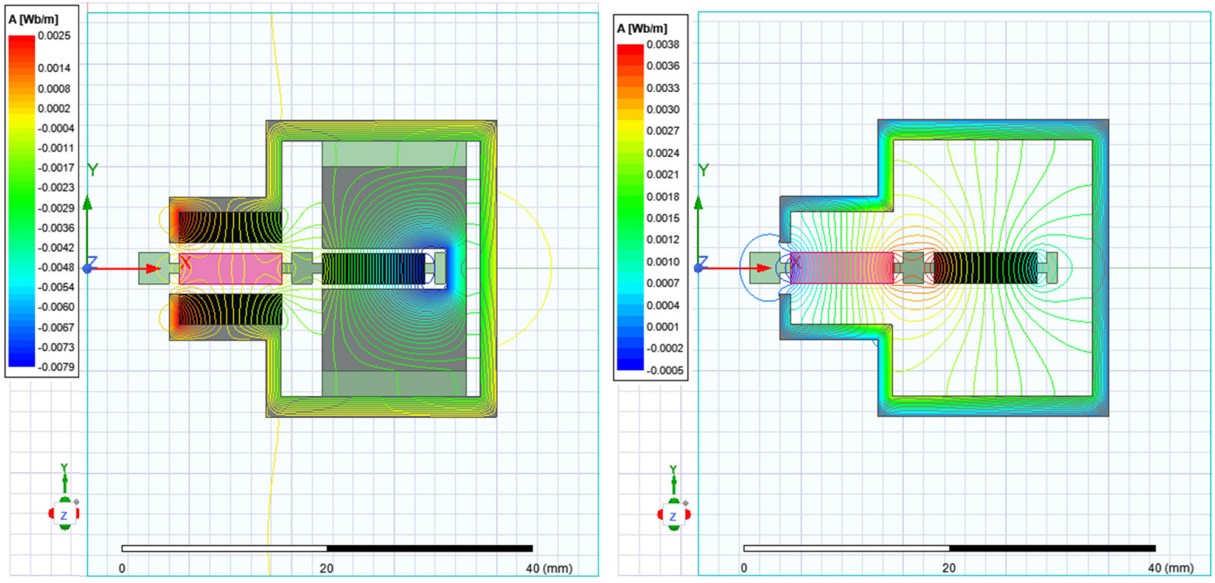

**Figure 7.** Magnetic flux lines distribution in stable positions (e.g., angular position = 0 degrees) and half the angular distance between two adjacent stable position (e.g., angular position = 22.5 degrees).

It can be observed that magnetic flux leakage is relatively small outside the envelope of the actuator. In addition, it is also minority inside the envelope, with some limited flux flowing between the stabilizer and the external yoke of the device and between the stator PMs and the external PM of the rotor.

On the other hand, a transient simulation model has been generated to obtain the electromagnetic losses induced by eddy currents and hysteresis. This FEM simulates the rotation of the rotor at a constant speed during a period of not less than 100 ms at intervals of 1 ms. Power losses are then calculated, at different rotor velocities, using the Steinmetz dynamic equation [23]:

$$P_v = P_h + P_c + P_e = k_h f B_m^2 + k_c (f B_m)^2 + k_e (f B_m)^{1.5} \tag{18}$$

where $P_v$ is the total specific core losses, $P_h$ is the hystresis core losses, $P_c$ the eddy current power losses, and $P_e$ are the excess power losses. Consequently, $k_h$ is the hysteresis coefficient, $k_c$ the eddy current coefficient and $k_e$ the excess coefficient. Finally, f denotes frequency, $B_m$ is the amplitude of the AC component of the magnetic flux density.

Figure 8 shows the specific power losses over a random instant of time when the rotor spins at 500 rpm. The structural parts have been removed in Figure 8 for a better visualization of the specific power losses; however, they contribute to the calculation of the system power losses.

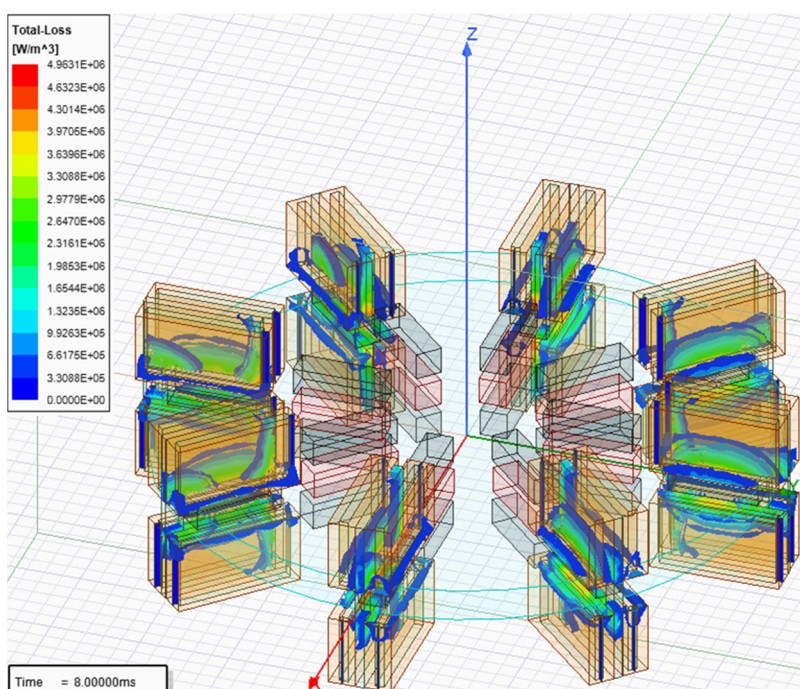

**Figure 8.** Specific magnetic loss distribution during an instant of the transient simulation.

The properties of the materials considered in the simulations are reported in Table 3. AISI 304 has been considered a perfect austenitic alloy. Although deformations and heat treatment processes may induce some magnetization effects [24], they are considered negligible in this study. In addition, the magnetic wire has been considered to be made of high-purity copper, and to be perfectly paramagnetic.

**Table 3.** Main properties of the materials considered in the model for transient and magnetostatic simulations. [24,25].

| Material | Component | Properties | | |
|---|---|---|---|---|
| | | **Electrical Conductivity [MS/m]** | **Magnetic Saturation J [T] or Remanence Br [T]** | **Coercitivity Hcb [A/m]** |
| AISI 304 | Rotor | 1.4 | $\approx 0$ | $\approx 0$ |
| AISI 420 | Stator | 1.82 | J = 1.45 | 800 |
| AISI 1010 | Stabilizer & coil cores | 7 | J = 2.09 | 635 |
| Copper | Magnetic wire | 58.8 | N/A | N/A |
| Sm Co$_{17}$ | Magnets | 1.82 | Br = 1.12 | 835,000 |

## 6. Performance Results

This section presents the theoretical calculations supported by the results obtained through the finite element analysis (FEA) models detailed in the previous section.

### 6.1. Actuation Torque and Stability of the Equilibrium Positions

The torque exerted on the rotor vs. its angular position is calculated using the FEA magnetostatic model. Results are reported in Figure 9.

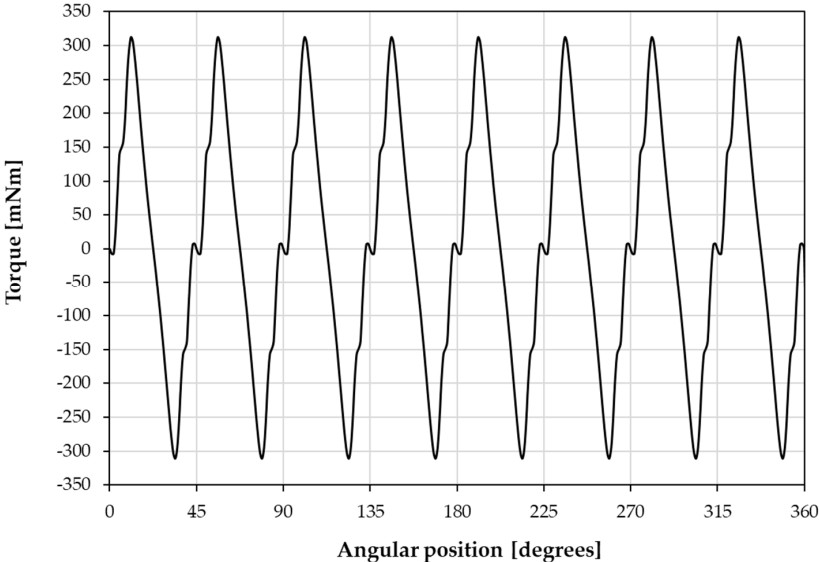

**Figure 9.** Torque vs. rotor angular position obtained via FEM calculations.

It is observed that the maximum actuation torque is 312 mNm. It must be noted that the magnetic potential energy and the kinetic energy of the system are related to the previous result. Moreover, the size of the different magnetic elements in the presented design is susceptible to optimization, for example, in order to maximize the torque, minimize the PM material, or minimize the device weight. This optimization could be achieved using the presented FEM. In addition, using more complex Halbach permanent magnet arrays could also lead to a better energy density of the device. However, such optimization is out of the scope of the present paper. Eight equilibrium positions are observed at $\theta_{eq} = 0 + 45° \cdot (n - 1)$, where n represents the identification number of each equilibrium position and can be any integer number between 1 and 8. The zone of stability has an amplitude of approximately ±2.3 degrees around those equilibrium positions.

The detent torque of the actuator is about 10 mNm. Figure 10 shows a detail of the torque vs. position curve around a stable equilibrium position. The accuracy in the position of the actuator can be interpreted from these results. It can be said that the rotor will present an accuracy better than ±2.3 degrees around the stable positions under normal operating conditions. The exact position of the rotor would depend on the presence of external static torques, disturbances, or energy-dissipative phenomena.

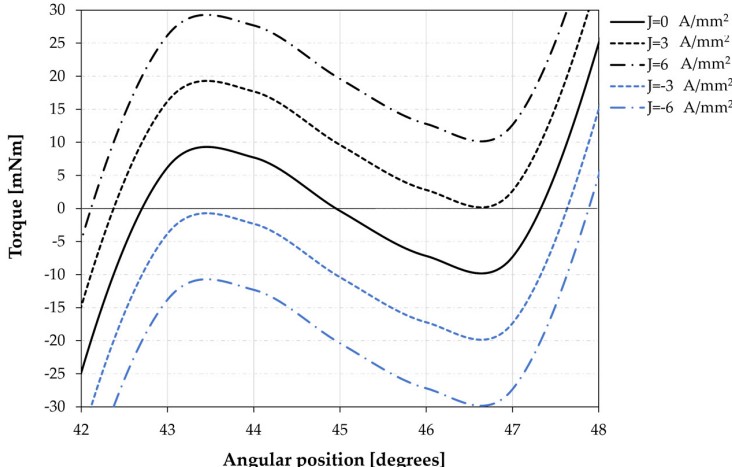

**Figure 10.** Influence of the current in the coils as a function of the rotor position obtained via FEM calculations.

To move from one stability position to another, it will be necessary to overcome this potential energy well. Through the integration of the area of the curve in Figure 10, it can be calculated that the minimum required energy to overcome the stabilization effect is about 13 ± 1 mJ. This energy would be supplied by the actuation system to generate a position change. Figure 10 shows the influence of the input current density on the stability zone. See that the direction of the circulating current establishes the direction of rotation of the actuator. For positive current densities, the actuator tends to rotate counterclockwise (positive), while the actuator will rotate clockwise (negative) if the supplied current is negative.

It can be seen that for current densities higher than 3 A/mm$^2$, the rotor has sufficient energy to overcome the stability zone and initiate rotation to the next stable position. Note that these results do not consider the action of a static friction pair on the bearings. Such a friction torque would cause a displacement in the curves described in Figure 10; therefore, a slightly higher current level would be necessary. Considering the characteristics of the winding described in the third paragraph, an actuation current density of 3.5 A/mm$^2$ is defined. Given the characteristics of the magnetic wire, the current maximum current consumption is estimated in 0.37 A. Results in Figure 10 are reported for both windings actuating at the same time. Operation in the redundant mode previously described would require about twice the current density.

### 6.2. Speed and Position Change Time

Figure 11 shows the profile of angular velocities in an event of position change between two equilibrium positions when the system has been actuated with a current of 3.5 A/mm$^2$. The speed increases until reaching its maximum at about 22.5 degrees from the stable position, its maximum value being about 700 rpm. From that moment, the system begins to decelerate, braking completely in the next stability zone.

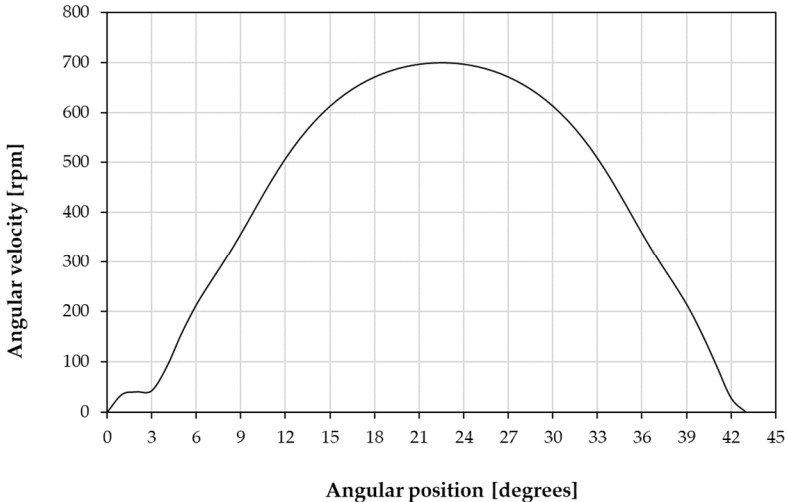

**Figure 11.** Angular velocity and acceleration vs. angle.

Using Equation (7) and from the simulated data, the angular position profile with respect to time can be obtained. The minimum time of position change is calculated at about 56.4 ms for the device operating without an attached payload. Figure 12 shows the relationship between the moment of inertia of the payload and the moment of inertia of the rotor with the time of change of position and the peak speed of rotation.

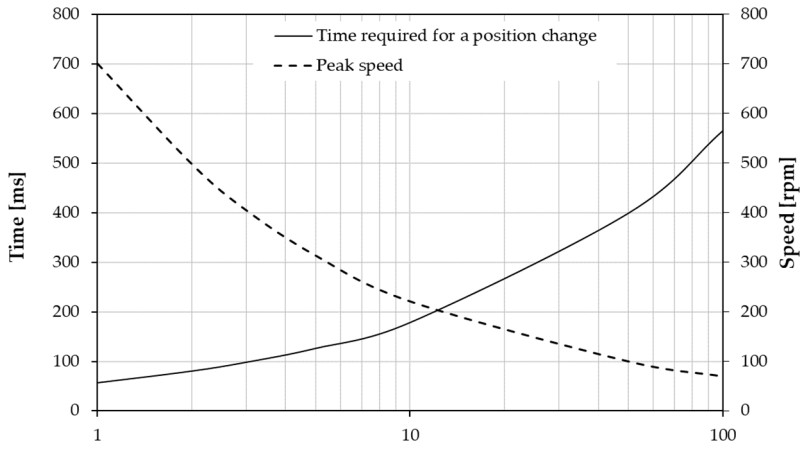

**Figure 12.** Average speed and time of change of position vs. ratio of moments of inertia.

*6.3. Power and Energy Consumption*

An AWG 28 electric wire has been selected for the construction of the coils. The maximum temperature of this cable is considered equal to the defined limit of operating temperatures of the device of 70 °C. In such conditions, the resistance of the windings, and therefore the power consumption of the device, is maximum. In addition, it has been considered that, due to the intermittent nature of the operation windings and the short periods of action, the winding temperature does not increase significantly during operation. The total estimated length of the winding is 44 m. Therefore, using Equation (10), a maximum winding resistance of 11 Ohm is calculated at an operational temperature of 70 °C. The expected maximum consumption of the actuation system ($W_{act}$) is then calculated in about 1.5 W. Figure 13 shows the absolute value of the apparent power ($W_{apparent}$) and the power consumed by the device as a function of its angular position.

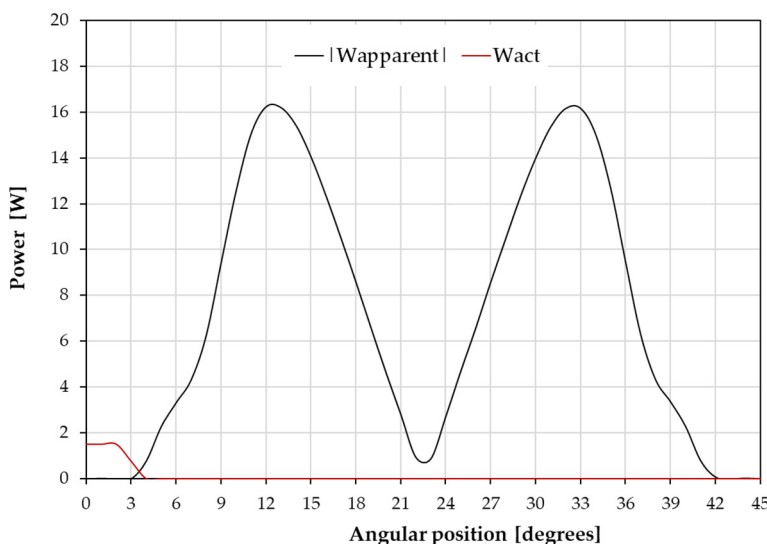

**Figure 13.** Instantaneous power consumption vs. rotor position.

By comparing the maximum value of the apparent power (16.2 W) with the value of the power consumed by the actuator (1.5 W), an ideal power saving can be defined:

$$\eta_{W\_ideal} = 90.7\ \%$$

However, other sources of energy dissipation must be considered when evaluating the behavior of the actuator. The two main sources of energy dissipation are:

- Friction in the bearings;
- Induced magnetic losses.

Magnetic losses induced by hysteresis and eddy currents in actuator materials have been calculated using the transient FEM described in Section 4. Friction losses in bearings have been calculated considering a friction torque similar to that of the James Webb Space Telescope's NIRCam instrument bearings. A maximum friction torque of 9.6 mNm is defined [6]. Figure 14 shows the power dissipated due to these losses as a function of the rotational speed of the rotor.

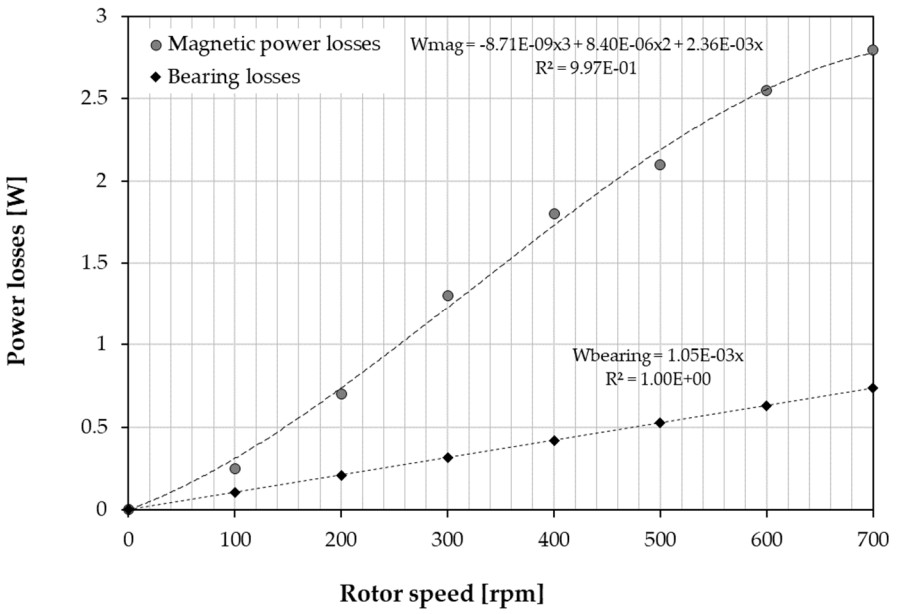

**Figure 14.** Magnetic power losses (FEM) and bearing losses (analytical) vs. rotor angular velocity.

A time-averaged actuation speed of 133 rpm is defined. This velocity is considered for the average power dissipation due to magnetic and friction losses. Then:

$$\overline{W}_{mag} = 0.68 \text{ W and } \overline{W}_{rod} = 0.09 \text{ W}$$

And

$$\overline{W}_{los} = \overline{W}_{mag} + \overline{W}_{rod} = 0.77 \text{ W}$$

Therefore, the power-saving ratio is calculated using Equation (16) as:

$$\eta_{W\_los}[\%] = 86.6 \text{ %}$$

From these values of average power consumption, and considering a position time change of 56.4 ms, the dissipative energy due to power losses can be calculated.

$$E_{los} = \overline{W}_{los} \cdot \Delta t = 43.4 \text{ } mJ$$

The actuation energy is calculated considering the power consumption of the windings (1.5 W) and the actuation time, which is calculated at about 15 ms using Equation (7). In addition, the apparent energy consumption is calculated using Equation (14) with data from Figure 10:

$$E_{act} \approx 23mJ \text{ and } E_{ap} \approx 117 \text{ } mJ$$

Finally, the estimated energy-saving ratio is calculated using Equation (17):

$$\eta_{E\_los}[\%] = 58,6 \text{ %}$$

When these results are compared to the performance of the actuators reported in Table 1, it can be appreciated that the proposed novel actuator concept has the potential to provide a superior actuation torque with less power and energy consumption. In addition, the time for position change is significantly reduced.

### 7. Conclusions

In this paper, a novel concept of a rotary electromagnetic actuator for positioning of a payload based on the harnessing of potential magnetic energy is presented. Its operational principle is discussed and a theoretical model for its performance calculation is presented. A prototype for potential application as an actuator in filter wheel applications is designed with an outer diameter of 92 mm and 40 mm in length. The device includes eight equilibrium positions and a single-phase redundant winding for position change. The performance of the prototype has been evaluated by a combination of the theoretical model described in this paper and both magnetostatic and transient FEM results.

A maximum actuation torque of about 312 mNm and a detent torque of about 10 mNm have been calculated. By applying an actuation current density in the windings of the actuator of about 3.5 A/mm$^2$, the rotor is able to change from one stable position to another. The time required for position change is estimated at 56.4 ms. In addition, the dependency of the position time change and the rotor speed with the momentum of inertia of the payload has been described.

Thanks to the harnessing of the potential magnetic energy stored in the equilibrium positions, a very significant power-saving ratio of up to 86.6% is calculated with regard to a classical electromagnetic actuator operating in the same conditions. Finally, an energy-saving ratio of up to 58.6% has been calculated. These promising results support the idea that the novel actuator concept could lead to very significant power and energy savings in applications demanding fast and accurate positioning which currently use classical electromagnetic actuators, such as optical filter wheels.

**Author Contributions:** Software, M.A.-C. and M.F.-M.; Validation, D.L.-P.; Formal analysis, G.V.-A.; Investigation, M.A.-C.; Supervision, I.V.-B.; Project administration, I.V.-B.; Funding acquisition, I.V.-B. All authors have read and agreed to the published version of the manuscript.

**Funding:** This work has received funding from the Spanish Ministry of Science, Innovation and Universities under grant agreement number RYC-2017-23684.

**Data Availability Statement:** Data available on request.

**Conflicts of Interest:** The authors declare no conflict of interest.

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
