# Peer review of "A Novel Ultra-Low Power Consumption Electromagnetic Actuator Based on Potential Magnetic Energy: Theoretical and Finite Element Analysis"

_actuators, doi:10.3390/act12020087_

Round 1

Reviewer 1 Report

1.       A novel concept of a rotary electromagnetic actuator for positioning with ultra-low power consumption is presented. The topic is interesting. The experiments should be carried to verify the practical performance of the proposed actuator.

2.       Fig.3 illustrates the structure of VALMAC actuator. But the details of stabilization and actuation system are shadowy. I can not understand the structure well. The positional relation and magnetization direction of stabilizers and permanent magnets are also difficult to understand in Figs.3, 4, and 5.

3.       With energy dissipation, such as eddy currents dissipation or friction in the bearing, whether the rotor can be accurately positioned? And how to achieve precise control? Have you conducted relevant experiments?

4.       Can the mathematical model of torque and angular velocity be established?

Author Response

  1.  A novel concept of a rotary electromagnetic actuator for positioning with ultra-low power consumption is presented. The topic is interesting. The experiments should be carried to verify the practical performance of the proposed actuator.

 Thanks for appreciating the novelty of the proposed device. However, to provide an experimental validation of the device is totally out of the scope of the paper, which is strictly based in the theoretical modelling of the conceptual design. The theoretical formulation is solid and is validate by FEA, which we consider is sufficient at this stage of the research. Even though experimental validation would be an ideal conclusion to this research, it is not possible at this stage due to economic and time limitations. However, sufficient details are provided for any external researcher to reproduce our work or even manufacture and test the device themselves.

In addition, we have modified the title of the paper to “A novel ultra-low power consumption electromagnetic actuator based on potential magnetic energy: theoretical and finite element analysis” in order to be clear with the scope of the work.

Please, reconsider accepting the paper as strictly theoretical (as many many others in the literature).

  1. Fig.3 illustrates the structure of VALMAC actuator. But the details of stabilization and actuation system are shadowy. I cannot understand the structure well. The positional relation and magnetization direction of stabilizers and permanent magnets are also difficult to understand in Figs. 3, 4, and 5.

Agreed, we have included a new figure (Figure 4) giving further assembly details. Please, also consider that the magnetization directions, distribution of the elements and dimensions are already described in the text of the paper immediately before Fig. 4. In addition, further details are now provided on the magnetic circuit arrangement in Fig. 6.

  1. With energy dissipation, such as eddy currents dissipation or friction in the bearing, whether the rotor can be accurately positioned? And how to achieve precise control? Have you conducted relevant experiments?

This a very interesting comment. Any energy dissipation phenomena would impact the positioning of the device. As energy is extracted from the system, it should be added again in order to assure that the final position of the rotor meets the stabilization zone. Once within the stabilization zone, hysteresis dissipation on the ferromagnetic material would also have an impact on the final position of the rotor. This would be corrected and controlled by an adequate control strategy based in the measurement of the position of the rotor. The integration of a set of hall effect sensor for indirect measurement of the position of the rotor is foreseen in a potential prototype of the technology. With this sensor, a close-loop feedback control can be implemented considering either position or speed control.

Despite a detailed discussion is out of the scope of the paper, we have included a paragraph at the very end of section 3 opening the discussion.  
            “Furthermore, the presence of this energy dissipation phenomena would reduce inexorably the kinetic energy of the rotor, potentially preventing it to achieve the next stable position. In order to overcome this limitation, as discussed in section 2, the actuation system provides the extra energy to compensate those losses. By measuring position and velocity of the rotor, a feedback control strategy could be potentially defined in order to pro-vide an accurate control of positioning of the actuator. Although the definition of this control strategy is out of the scope of this paper, a set of hall effect sensors to be used with control purposes have been foresee in the design of the actuator in section 4”.

  1.  Can the mathematical model of torque and angular velocity be established?

We understand that you refer to a mathematical model that links the electromagnetic interactions between the components of the system to obtain a formulation of the torque and velocity of the rotor. This should be possible; however, the formulation would be extremely complex and not possible to be solved analytically due to the non-linearity functions describing the magnetic interactions and the many different elements involved. Therefore, we have decided to solve the problem using FEA using a very well-demonstrated software tool such as ANSYS Electronics. This is described in section 3 lines 170 to 174. Therefore, obtaining torque vs. angular position of the rotor by FEA, the angular velocity of the system can be solved using eq. 6.

Reviewer 2 Report

General and minor comments: 1.      In the title is ”..Based in Potential..”, better is ..Based on Potential 2.      The paper gives a new conception of the low-power electromagnetic actuator for positioning. The declared power consumption saving of up to 86.6% and an energy consumption reduction of up to 58.6% should be justified and with which type and model of the traditional actuator are respected? 3.      The principle of operation is not clear regarding Fig. 1, there is probably a mistake with the labels of Rotor and Stator, I suppose that the rotor is the part with four magnets the while stator is the part with coils. 4.      According to the principle, potential energy is accumulated, when the rotor position is on the x-axis, not on the y-axis or on both axes. 5.      The potential energy and as well kinetic energy depend on the size of the magnets. Moreover, the consumed electric energy to supply coils/stabilizer is also strictly dependent, so how the size of the magnets are calculated and optimized? 6.      Paper provides some analytical and FEM designs, but there is a lack of calculations and mechanical design of the system elements and optimization. 7.      Some similar solutions provided for energy harvesting were published before, that should be referenced, i.e. a.       Design of vibration exciter by using permanent magnets for application to piezoelectric energy harvesting     Yang, C.H., Song, D., Woo, M.S., (...), Baek, K.H., Sung, T.H.      2012       Proceedings of 2012 21st IEEE Int. Symp. on Applications of Ferroelectrics held jointly with 11th IEEE European Conference on the Applications of Polar Dielectrics and IEEE PFM, ISAF/ECAPD/PFM 20126297748 b.      Experimental study of macro fiber composite-magnet energy harvester for self-powered active magnetic bearing rotor vibration sensor, Open Access, Mystkowski, A., Ostasevicius, V.  2020       Energies, 13(18),4806   8.      According to the magnet setups, better energy density can be obtained by using the Halbach arrays. 9.      The magnetic energy loss calculations are missed. 10.  The FFM analysis is given in a sloppy manner, it should be improved according to the magnetic and electromagnetic energy generated with magnets and coils for the rotor positions and must be done carefully in a separate section. 11.  The magnetic field distribution is presented poorly. 12.  Also, is not clear what are the simulation data and FFE model data. 13.  The flux losses analysis using FFM is missed. 14.  The actuator structure and mechanical/electrical data are missing. 15.  The system set-ups need more details.   I invite Authors to respond to the reviewer’s comments and revise their papers.

Author Response

Dear Reviewer, please find answer to your comments in the attached file

Round 2

Reviewer 1 Report

No comments

Reviewer 2 Report

no more comments